# The expressive power of pooling in Graph Neural Networks

**Filippo Maria Bianchi**[*]
Dept. of Mathematics and Statistics
UiT the Arctic University of Norway
NORCE, Norwegian Research Centre AS
`filippo.m.bianchi@uit.no`

**Veronica Lachi**[*]
Dept. of Information Engineering and Mathematics
University of Siena
`veronica.lachi@student.unisi.it`

## Abstract

In Graph Neural Networks (GNNs), hierarchical pooling operators generate local summaries of the data by coarsening the graph structure and the vertex features. While considerable attention has been devoted to analyzing the expressive power of message-passing (MP) layers in GNNs, a study on how graph pooling affects the expressiveness of a GNN is still lacking. Additionally, despite the recent advances in the design of pooling operators, there is not a principled criterion to compare them. In this work, we derive sufficient conditions for a pooling operator to fully preserve the expressive power of the MP layers before it. These conditions serve as a universal and theoretically grounded criterion for choosing among existing pooling operators or designing new ones. Based on our theoretical findings, we analyze several existing pooling operators and identify those that fail to satisfy the expressiveness conditions. Finally, we introduce an experimental setup to verify empirically the expressive power of a GNN equipped with pooling layers, in terms of its capability to perform a graph isomorphism test.

## 1 Introduction

Significant effort has been devoted to characterizing the expressive power of Graph Neural Networks (GNNs) in terms of their capabilities for testing graph isomorphism [34]. This has led to a better understanding of the strengths and weaknesses of GNNs and opened up new avenues for developing advanced GNN models that go beyond the limitations of such algorithms [32]. The more powerful a GNN, the larger the set of non-isomorphic graphs that it can distinguish by generating distinct representations for them. GNNs with appropriately formulated message-passing (MP) layers are as effective as the Weisfeiler-Lehman isomorphism test (WL test) in distinguish graphs [38], while higher-order GNN architectures can match the expressiveness of the $k$-WL test [29]. Several approaches have been developed to enhance the expressive power of GNNs by incorporating random features into the nodes [33, 1], by using randomized weights in the network architecture [42], or by using compositions of invariant and equivariant functions [27]. Despite the progress made in understanding the expressive power of GNNs, the results are still limited to *flat* GNNs consisting of a stack of MP layers followed by a final readout [38, 5].

Inspired by pooling in convolutional neural networks, recent works introduced hierarchical pooling operators that enable GNNs to learn increasingly abstract and coarser representations of the input graphs [39, 9]. By interleaving MP with pooling layers that gradually distill global graph properties through the computation of local summaries, it is possible to build deep GNNs that improve the accuracy in graph classification [8, 4] and node classification tasks [18, 26].

---

[*]Equal contribution

37th Conference on Neural Information Processing Systems (NeurIPS 2023).

It is not straightforward to evaluate the power of a graph pooling operator and the quality of the coarsened graphs it produces. The most common approach is to simply measure the performance of a GNN with pooling layers on a downstream task, such as graph classification. However, such an approach is highly empirical and provides an indirect evaluation affected by external factors. One factor is the overall GNN architecture: pooling is combined with different MP layers, activation functions, normalization or dropout layers, and optimization algorithms, which makes it difficult to disentangle the contribution of the individual components. Another factor is the dataset at hand: some classification tasks only require isolating a specific motif in the graph [23, 7], while others require considering global properties that depend on the whole graph structure [15]. Recently, two criteria were proposed to evaluate a graph pooling operator in terms of i) the spectral similarity between the original and the coarsened graph topology and ii) its capability of reconstructing the features of the original graph from the coarsened one [20]. While providing valuable insights, these criteria give results that are, to some extent, contrasting and in disagreement with the traditional evaluation based on the performance of the downstream task.

To address this issue, we introduce a universal and principled criterion that quantifies the power of a pooling operator as its capability to retain the information in the graph from an expressiveness perspective. In particular, we investigate how graph pooling affects the expressive power of GNNs and derive sufficient conditions under which the pooling operator preserves the highest degree of expressiveness. Our contributions are summarized as follows.

- We show that when certain conditions are met in the MP layers and in the pooling operator, their combination produces an injective function between graphs. This implies that the GNN can effectively coarsen the graph to learn high-level data descriptors, without compromising its expressive power.

- Based on our theoretical analysis, we identify commonly used pooling operators that do not satisfy these conditions and may lead to failures in certain scenarios.

- We introduce a simple yet effective experimental setup for measuring, empirically, the expressive power of *any* GNN in terms of its capability to perform a graph isomorphism test.

Besides providing a criterion for choosing among existing pooling operators and for designing new ones, our findings allow us to debunk criticism and misconceptions about graph pooling.

## 2   Background

### 2.1   Graph neural networks

Let $\mathcal{G} = (\mathcal{V}, \mathcal{E})$ be a graph with node features $\mathbf{X}^0 \in \mathbb{R}^{N \times F}$, where $|\mathcal{V}| = N$. Each row $\mathbf{x}_i^0 \in \mathbb{R}^F$ of the matrix $\mathbf{X}^0$ represents the initial node feature of the node $i$, $\forall i = 1, \dots, N$. Through the MP layers a GNN implements a local computational mechanism to process graphs [19]. Specifically, each feature vector $\mathbf{x}_v$ is updated by combining the features of the neighboring nodes. After $l$ iterations, $\mathbf{x}_v^l$ embeds both the structural information and the content of the nodes in the $l$–hop neighborhood of $v$. With enough iterations, the feature vectors can be used to classify the nodes or the entire graph. More rigorously, the output of the $l$-th layer of a MP-GNN is:

$$\mathbf{x}_v^l = \texttt{COMBINE}^{(l)}(\mathbf{x}_v^{l-1}, \texttt{AGGREGATE}^{(l)}(\{\mathbf{x}_u^{l-1},\ u \in \mathcal{N}[v]\})) \qquad (1)$$

where $\texttt{AGGREGATE}^{(l)}$ is a function that aggregates the node features from the neighborhood $\mathcal{N}[v]$ at the $(l-1)$–th iteration, and $\texttt{COMBINE}^{(l)}$ is a function that combines the own features with those of the neighbors. This type of MP-GNN implements permutation-invariant feature aggregation functions and the information propagation is isotropic [35]. In graph classification/regression tasks, a $\texttt{READOUT}$ function typically transforms the feature vectors from the last layer $L$ to produce the final output:

$$\mathbf{o}\ =\ \texttt{READOUT}(\{\mathbf{x}_v^L,\ v \in \mathcal{V}\}). \qquad (2)$$

The $\texttt{READOUT}$ is implemented as the sum, mean, or the maximum of all node features, or by more elaborated functions [11, 40, 22].

## 2.2 Expressive power of graph neural networks

When analyzing the expressive power of GNNs, the primary objective is to evaluate their capacity to produce different outputs for non-isomorphic graphs. While an exact test for graph isomorphism has a combinatorial complexity [2], the WL test for graph isomorphism [37] is a computationally efficient yet effective test that can distinguish a broad range of graphs. The algorithm assigns to each graph vertex a color that depends on the multiset of labels of its neighbors and on its own color. At each iteration, the colors of the vertices are updated until convergence is reached.

There is a strong analogy between an iteration of the WL-test and the aggregation scheme implemented by MP in GNNs. In fact, it has been proven that MP-GNNs are at most as powerful as the WL test in distinguishing different graph-structured features [38, 29]. Moreover, if the MP operation is injective, the resulting MP-GNN is as powerful as the WL test [38]. The Graph Isomorphism Network (GIN) implements such an injective multiset function as:

$$\mathbf{x}_v^l = \texttt{MLP}^{(l)} \left( (1 + \epsilon^l)\mathbf{x}_v^{l-1} + \sum_{u \in \mathcal{N}[v]} \mathbf{x}_u^{l-1} \right). \tag{3}$$

Under the condition that the nodes' features are from a countable multiset, the representational power of GIN equals that of the WL test. Some GNNs can surpass the discriminative power of the WL test by using higher-order generalizations of MP operation [29], or by using a composition of invariant and equivariant functions [27], at the price of higher computational complexity. In this work, we focus on the standard MP-GNN, which remains the most widely adopted due to its computational efficiency.

## 2.3 Graph pooling operators

A graph pooling operator implements a function $\texttt{POOL} : \mathcal{G} \mapsto \mathcal{G}_P = (\mathcal{V}_P, \mathcal{E}_P)$ such that $|\mathcal{V}_P| = K$, with $K \leq N$. We let $\mathbf{X}_P \in \mathbb{R}^{K \times F}$ be the pooled nodes features, i.e., the features of the nodes $\mathcal{V}_P$ in the pooled graph. To formally describe the $\texttt{POOL}$ function, we adopt the Select-Reduce-Connect (SRC) framework [20], that expresses a graph pooling operator through the combination of three functions: *selection*, *reduction*, and *connection*. The selection function (SEL) clusters the nodes of the input graph into subsets called *supernodes*, namely $\texttt{SEL} : \mathcal{G} \mapsto \mathcal{S} = \{\mathcal{S}_1, \ldots, \mathcal{S}_K\}$ with $\mathcal{S}_j = \left\{ s_i^j \right\}_{i=1}^N$ where $s_i^j$ is the membership score of node $i$ to supernode $j$. The memberships are conveniently represented by a cluster assignment matrix $\mathbf{S}$, with entries $[\mathbf{S}]_{ij} = s_i^j$. Typically, a node can be assigned to zero, one, or several supernodes, each with different scores. The reduction function (RED) creates the pooled vertex features by aggregating the features of the vertices assigned to the same supernode, that is, $\texttt{RED} : (\mathcal{G}, \mathbf{S}) \mapsto \mathbf{X}_P$. Finally, the connect function (CON) generates the edges, and potentially the edge features, by connecting the supernodes.

# 3 Expressive power of graph pooling operators

We define the expressive power of a graph pooling operator as its capability of preserving the expressive power of the MP layers that came before it. We first present our main result, which is a formal criterion to determine the expressive power of a pooling operator. In particular, we provide three sufficient (though not necessary) conditions ensuring that if the MP and the pooling layers meet certain criteria, then the latter retains the same level of expressive power as the former. Then, we analyze several existing pooling operators and analyze their expressive power based on those criteria.

## 3.1 Conditions for expressiveness

**Theorem 1.** *Let $\mathcal{G}_1 = (\mathcal{V}_1, \mathcal{E}_1)$ with $|\mathcal{V}_1| = N$ and $\mathcal{G}_2 = (\mathcal{V}_2, \mathcal{E}_2)$ with $|\mathcal{V}_2| = M$ with node features $\mathbf{X}$ and $\mathbf{Y}$ respectively, such that $\mathcal{G}_1 \neq_{WL} \mathcal{G}_2$. Let $\mathcal{G}_1^L$ and $\mathcal{G}_2^L$ be the graph obtained after applying a block of $L$ MP layers such that $\mathbf{X}^L \in \mathbb{R}^{N \times F}$ and $\mathbf{Y}^L \in \mathbb{R}^{M \times F}$ are the new nodes features. Let $\texttt{POOL}$ be a pooling operator expressed by the functions $\texttt{SEL}, \texttt{RED}, \texttt{CON}$, which is placed after the MP layers. Let $\mathcal{G}_{1_P} = \texttt{POOL}(\mathcal{G}_1^L)$ and $\mathcal{G}_{2_P} = \texttt{POOL}(\mathcal{G}_2^L)$ with $|\mathcal{V}_{1_P}| = |\mathcal{V}_{2_P}| = K$. Let $\mathbf{X}_P \in \mathbb{R}^{K \times F}$ and $\mathbf{Y}_P \in \mathbb{R}^{K \times F}$ be the nodes features of the pooled graphs so that the rows $\mathbf{x}_{P_j}$ and $\mathbf{y}_{P_j}$ represent the features of supernode $j$ in graphs $\mathcal{G}_{1_P}$ and $\mathcal{G}_{2_P}$, respectively. If the following conditions hold:*

1. $\sum_i^N \mathbf{x}_i^L \neq \sum_i^M \mathbf{y}_i^L$;

2. *The memberships generated by* `SEL` *satisfy* $\sum_{j=1}^K s_{ij} = \lambda$, *with* $\lambda > 0$ *for each node* $i$, *i.e., the cluster assignment matrix* $\mathbf{S}$ *is a right stochastic matrix up to the global constant* $\lambda$;

3. *The function* `RED` *is of type* `RED` $: (\mathbf{X}^L, \mathbf{S}) \mapsto \mathbf{X}_P = \mathbf{S}^T \mathbf{X}^L$;

*then* $\mathcal{G}_{1_P}$ *and* $\mathcal{G}_{2_P}$ *will have different nodes features, i.e., for all rows' indices permutations* $\pi :$ $\{1, \ldots K\} \to \{1, \ldots K\}$, $\mathbf{X}_P \neq \Pi(\mathbf{Y}_P)$, *where* $[\Pi(\mathbf{Y}_P)]_{ij} = \mathbf{y}_{P_{\pi(i)j}}$.

The proof can be found in Appendix A and a schematic summary is in Fig. 1.

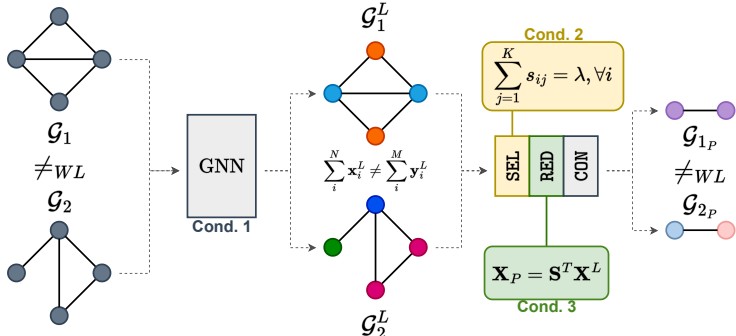

Figure 1: A GNN with expressive MP layers (condition 1) computes different features $\mathcal{X}_1^L$ and $\mathcal{X}_2^L$ for two graphs $\mathcal{G}_1, \mathcal{G}_2$ that are WL-distinguishable. A pooling layer satisfying the conditions 2 and 3 generates coarsened graphs $\mathcal{G}_{1_P}$ and $\mathcal{G}_{2_P}$ that are still WL-distinguishable.

Condition 1 is strictly related to the theory of multisets. Indeed, a major breakthrough in designing highly expressive MP layers was achieved by building upon the findings of Deep Sets [41]. Under the assumption that the node features originate from a countable universe, it has been formally proven that there exists a function that, when applied to the node features, makes the sum over a multiset of node features injective [38]. The universal approximation theorem guarantees that such a function can be implemented by an MLP. Moreover, if the pooling operator satisfies conditions 2 and 3, it will produce different sets of node features. Due to the injectiveness of the coloring function of the WL algorithm, two graphs with different multisets of node features will be classified as non-isomorphic by the WL test and, therefore, $\mathcal{G}_{1_P} \neq_{\text{WL}} \mathcal{G}_{2_P}$. This implies that the pooling operator effectively coarsens the graphs while retaining all the information necessary to distinguish them. Therefore, our Theorem ensures that there exists a specific choice of parameters for the MP layer that, when combined with a pooling operator satisfying the Theorem's conditions, the resulting GNN architecture is injective.

Condition 2 implies that *all* nodes in the original graph must contribute to the supernodes. Moreover, letting the sum of the memberships $s_{ij}$ to be a constant $\lambda$ (usually, $\lambda = 1$), places a restriction on the formation of the super-nodes. Condition 3 requires that the features of the supernodes $\mathbf{X}_P$ are a convex combination of the node features $\mathbf{X}^L$. It is important to note that the conditions for the expressiveness only involve `SEL` and `RED`, but not the `CON` function. Indeed, both the graph's topology and the nodes' features are embedded in the features of the supernodes by MP and pooling layers satisfying the conditions of Th. 1. Nevertheless, even if a badly-behaved `CON` function does not affect the expressiveness of the pooling operator, it can still compromise the effectiveness of the MP layers that come afterward. This will be discussed further in Sections 3.3 and 4.

## 3.2 Expressiveness of existing pooling operators

The SRC framework allows building a comprehensive taxonomy of the existing pooling operators, based on the density of supernodes, the trainability of the `SEL`, `RED`, and `CON` functions, and the adaptability of the number of supernodes $K$ [20]. The density of a pooling operator is defined as the expected value of the ratio between the cardinality of a supernode and the number of nodes in the graph. A method is referred to as *dense* if the supernodes have cardinality $O(N)$, whereas a pooling operator is considered *sparse* if the supernodes generated have constant cardinality $O(1)$ [20].

Pooling methods can also be distinguished according to the number of nodes $K$ of the pooled graph. If $K$ is constant and independent of the input graph size, the pooling method is *fixed*. On the other hand, if the number of supernodes is a function of the input graph, the method is *adaptive*. Finally, in some pooling operators the `SEL`, `RED`, and `CON` functions can be learned end-to-end along with the other components of the GNN architecture. In this case, the method is said to be *trainable*, meaning that the operator has parameters that are learned by optimizing a task-driven loss function. Otherwise, the methods are *non-trainable*.

**Dense pooling operators** Prominent methods in this class of pooling operators are DiffPool [39], MinCutPool [8], and DMoN [36]. Besides being dense, all these operators are also trainable and fixed. DiffPool, MinCutPool, and DMoN compute a cluster assignment matrix $\mathbf{S} \in \mathbb{R}^{N \times K}$ either with an MLP or an MP-layer, which is fed with the node features $\mathbf{X}^L$ and ends with a `softmax`. The main difference among these methods is in how they define unsupervised auxiliary loss functions, which are used to inject a bias in how the clusters are formed. Thanks to the `softmax` normalization, the cluster assignments sum up to one, ensuring condition 2 of Th. 1 to be satisfied. Moreover, the pooled node features are computed as $\mathbf{X}_p = \mathbf{S}^{\mathrm{T}} \mathbf{X}^L$, making also condition 3 satisfied.

There are dense pooling operators that use algorithms such as non-negative matrix factorization [3] to obtain a cluster assignment matrix $\mathbf{S}$, which may not satisfy condition 2. Nonetheless, it is always possible to apply a suitable normalization to ensure that the rows in $\mathbf{S}$ sum up to a constant. Therefore, we claim that all dense methods preserve the expressive power of the preceding MP layers.

**Non-expressive sparse pooling operators** Members of this category are Top-$k$ [18, 23] AS-APool [31], SAGPool [24] and PanPool [26], which are also trainable and adaptive. These methods reduce the graph by selecting a subset of its nodes based on a ranking score and they mainly differ in how their `SEL` function computes such a score. Specifically, the Top-$k$ method ranks nodes based on a score obtained by multiplying the node features with a trainable projection vector. A node $i$ is kept ($s_i = 1$) if is among the top-$K$ in the ranking and is discarded ($s_i = 0$) otherwise. SAGPool simply replaces the projection vector with an MP layer to account for the graph's structure when scoring the nodes. ASAPool, instead, examines all potential local clusters in the input graph given a fixed receptive field and it employs an attention mechanism to compute the cluster membership of the nodes. The clusters are subsequently scored using a particular MP operator. Finally, in PanPool the scores are obtained from the diagonal entries of the maximal entropy transition matrix, which is a generalization of the graph Laplacian.

Regardless of how the score is computed, all these methods generate a cluster assignment matrix $\mathbf{S}$ where not all the rows sum to a constant. Indeed, if a node is not selected, it is not assigned to any supernode in the coarsened graph. Therefore, these methods fail to meet condition 2 of Theorem 1. Additionally, in the `RED` function of all these methods the features of each selected node are multiplied by its ranking score, making condition 3 also unsatisfied.

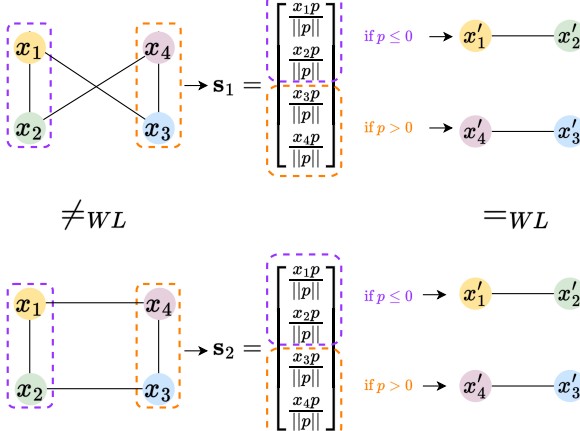

Figure 2: Example of failure of Top-$k$ pooling. Given two WL-distinguishable graphs with node features $x_1 \le x_2 \le x_3 \le x_4$, two scoring vectors $\mathbf{s}_1$ and $\mathbf{s}_2$ are computed using a projector $p$. Then, the two nodes associated with the highest scores are selected. If $p \le 0$, nodes 1 and 2 are chosen in both graphs. Conversely, if $p > 0$, nodes 3 and 4 are selected. Therefore, regardless of the value learned for the projector $p$, the two input graphs will be mapped into the same pooled graph.

Intuitively, these operators produce a pooled graph that is a subgraph of the original graph and discard the content of the remaining parts. This hinders the ability to retain all the necessary information for

preserving the expressiveness of the preceding MP layers. The limitation of Top-$k$ is exemplified in Fig. 2: regardless of the projector $p$, Top-$k$ maps two WL-distinguishable graphs into two isomorphic graphs, meaning that it cannot preserve the partition on graphs induced by the WL test.

**Expressive sparse pooling operators**  Not all sparse pooling operators coarsen the graph by selecting a subgraph. In fact, some of them assign each node in the original graph to exactly one supernode and, thus, satisfy condition 2 of Th. 1. In matrix form and letting $\lambda = 1$, the cluster assignment would be represented by a sparse matrix $\mathbf{S}$ that satisfies $\mathbf{S1}_K = \mathbf{1}_N$ and where every row has one entry equal to one and the others equal to zero. Within this category of sparse pooling operators, notable examples include Graclus [13], ECPool [14], and $k$-MISPool [4].

Graclus is a non-trainable, greedy bottom-up spectral clustering algorithm, which matches each vertex with the neighbor that is closest according to the graph connectivity [13]. When Graclus is used to perform graph pooling, the RED function is usually implemented as a `max_pool` operation between the vertices assigned to the same cluster [12]. In this work, to ensure condition 3 of Th. 1 to be satisfied, we use a `sum_pool` operation instead. Contrarily from Gralcus, ECPool, and $k$-MISPool are trainable. ECPool first assigns to each edge $e_{i \to j}$ a score $r_{ij} = f(\boldsymbol{x}_i, \boldsymbol{x}_j; \boldsymbol{\Theta})$. Then, iterates over each edge $e_{i \to j}$, starting from those with higher scores, and contracts it if neither nodes $i$ and $j$ are attached to an already contracted edge. The endpoints of a contracted edge are merged into a new supernode $\mathcal{S}_k = r_{ij}(\boldsymbol{x}_i + \boldsymbol{x}_j)$, while the remaining nodes become supernodes themselves. Since each supernode either contains the nodes of a contracted edge or is a node from the original graph, all columns of $\mathbf{S}$ have either one or two entries equal to one, while each row sums up to one. The RED function can be expressed as $\mathbf{r} \odot \mathbf{S}^T \mathbf{X}^L$, where $\mathbf{r}[k] = r_{ij}$ if $k$ is the contraction of two nodes $i$ $j$, otherwise $\mathbf{r}[k] = 1$. As a result, ECPool met the expressiveness conditions of Th. 1. Finally, $k$-MISPool identifies the supernodes with the centroids of the maximal $k$-independent sets of a graph [6]. To speed up computation, the centroids are selected with a greedy approach based on a ranking vector $\pi$. Since $\pi$ can be obtained from a trainable projector $\mathbf{p}$ applied to the vertex features, $\pi = \mathbf{X}^L \mathbf{p}^T$, $k$-MISPool is a trainable pooling operator. $k$-MISPool assigns each vertex to one of the centroids and aggregates the features of the vertex assigned to the same centroid with a `sum_pool` operation to create the features of the supernodes. Therefore, $k$-MISPool satisfies the expressiveness conditions of Th. 1.

A common characteristic of these methods is that the number of supernodes $K$ cannot be directly specified. Graclus and ECPool achieve a pooling ratio of approximately 0.5 by roughly reducing each time the graph size by 50%. On the other hand, $k$-MISPool can control the coarsening level by computing the maximal independent set from $\mathcal{G}^k$, which is the graph where each node of $\mathcal{G}$ is connected to its $k$-hop neighbors. As the value of $k$ increases, the pooling ratio decreases.

### 3.3 Criticism on graph pooling

Recently, the effectiveness of graph pooling has been questioned using as an argument a set of empirical results aimed at exposing the weaknesses of certain pooling operators [28]. The experiments showed that using a randomized cluster assignment matrix $\mathbf{S}$ (followed by a `softmax` normalization) gives comparable results to using the assignment matrices learned by Diffpool [39] and MinCut-Pool [8]. Similarly, applying Graclus [13] on the complementary graph would give a performance similar to using the original graph.

We identified potential pitfalls in the proposed evaluation, which considered only pooling operators that are expressive and that, even after being modified, retain their expressive power. Clearly, even if expressiveness ensures that all the information is preserved in the pooled graph, its structure is corrupted when using a randomized $\mathbf{S}$ or a complementary graph. This hinders the effectiveness of the MP layers that come after pooling, as their inductive biases no longer match the data structure they receive. Notably, this might not affect certain classification tasks where the goal is to detect small structures, such as a motif in a molecule graph [25, 21], that are already captured by the MP layers before pooling.

To address these limitations, first, we propose to corrupt a pooling operator that is not expressive. In particular, we design a Top-$k$ pooling operator where the nodes are ranked based on a score that is sampled from a Normal distribution rather than being produced by a trainable function of the vertex features. Second, we evaluate all the modified pooling operators in a setting where the MP layers after pooling are essential for the task and show that the performance drop is significant.

# 4   Experimental Results

To empirically confirm the theoretical results presented in Section 3, we designed a synthetic dataset that is specifically tailored to evaluate the expressive power of a GNN. We considered a GNN with MP layers interleaved with 10 different pooling operators: DiffPool [39], DMoN [36], MinCut [8], ECPool [14], Graclus, $k$-MISPool  [4], Top-$k$ [18], PanPool [26], ASAPool [31], and SAGPool [24]. For each pooling method, we used the implementation in Pytorch Geometric [17] with the default configuration. In addition, following the setup used to criticize the effectiveness of graph pooling [28], we considered the following pooling operators: Rand-Dense, a dense pooling operator where the cluster assignment is a normalized random matrix; Rand-Sparse, a sparse operator that ranks nodes based on a score sampled from a Normal distribution; Cmp-Graclus, an operator that applies the Graclus algorithm on the complement graph.

## 4.1   The EXPWL1 dataset

Our experiments aim at evaluating the expressive power of MP layers when combined with pooling layers.  However, existing real-world and synthetic benchmark datasets are unsuitable for this purpose as they are not specifically designed to relate the power of GNNs to that of the WL test. Recently, the EXP dataset was proposed to test the capability of special GNNs to achieve higher expressive power than the WL test [1], which, however, goes beyond the scope of our evaluation. Therefore, we introduce a modified version of EXP called EXPWL1, which comprises a collection of graphs $\{\mathcal{G}_1, \ldots, \mathcal{G}_N, \mathcal{H}_1, \ldots, \mathcal{H}_N\}$ representing propositional formulas that can be satisfiable or unsatisfiable. Each pair $(\mathcal{G}_i, \mathcal{H}_i)$ in EXPWL1 consists of two non-isomorphic graphs distinguishable by a WL test, which encode formulas with opposite SAT outcomes. Therefore, any GNN that has an expressive power equal to the WL test can distinguish them and achieve approximately 100% classification accuracy on the dataset. Compared to the original EXP dataset, we increased the size of the dataset to a total of 3000 graphs and we also increased the size of each graph from an average of 55 nodes to 76 nodes. This was done to make it possible to apply an aggressive pooling without being left with a trivial graph structure. The EXPWL1 dataset and the code to reproduce the experimental results are publicly available[2].

## 4.2   Experimental procedure

To empirically evaluate which pooling operator maintains the expressive power of the MP layers preceding it, we first identified a GNN architecture without pooling layers, which achieves approximately 100% accuracy on the EXPWL1. We tried different baselines (details in Appendix C.1) and we found that a GNN with three GIN layers [38] followed by a `global_sum_pool` reaches the desired accuracy. Then, we inserted a pooling layer between the 2$^{nd}$ and 3$^{rd}$ GIN layer, which performs an aggressive pooling by using a pooling ratio of 0.1 that reduces the graph size by 90%. The details of the GNN architectures are in Appendix C.2. Besides 0.1, we also considered additional pooling ratios and we reported the results in Appendix C.3. To ensure a fair comparison, when testing each method we shuffled the datasets and created 10 different train/validation/test splits using the same random seed. We trained each model on all splits for 500 epochs and reported the average training time and the average test accuracy obtained by the models that achieved the lowest loss on the validation set. To validate our experimental approach, we also measured the performance of the proposed GNN architecture equipped with the different pooling layers on popular benchmark datasets for graph classification [30, 10].

## 4.3   Experimental Results

Table 1 reports the performances of different pooling operators on EXPWL1. These results are consistent with our theoretical findings: pooling operators that satisfy the conditions of Th. 1 achieve the highest average accuracy and, despite the aggressive pooling, they retain all the necessary information to achieve the same performance of a GNN without a pooling layer.  On the other hand, non-expressive pooling operators achieve a significantly lower accuracy as they are not able to correctly distinguish all graphs.

---

[2]https://github.com/FilippoMB/The-expressive-power-of-pooling-in-GNNs

| Pooling | s/epoch | GIN layers | Pool Ratio | Test Acc | Expressive |
|---|---|---|---|---|---|
| *No-pool* | 0.33s | 3 | – | $99.3_{\pm 0.3}$ | ✓ |
| **DiffPool** | 0.69s | 2+1 | 0.1 | $97.0_{\pm 2.4}$ | ✓ |
| **DMoN** | 0.75s | 2+1 | 0.1 | $99.0_{\pm 0.7}$ | ✓ |
| **MinCut** | 0.72s | 2+1 | 0.1 | $98.8_{\pm 0.4}$ | ✓ |
| **ECPool** | 20.71s | 2+1 | 0.2 | $100.0_{\pm 0.0}$ | ✓ |
| **Graclus** | 1.00s | 2+1 | 0.1 | $99.9_{\pm 0.1}$ | ✓ |
| **$k$-MIS** | 1.17s | 2+1 | 0.1 | $99.9_{\pm 0.1}$ | ✓ |
| **Top-$k$** | 0.47s | 2+1 | 0.1 | $67.9_{\pm 13.9}$ | ✗ |
| **PanPool** | 3.82s | 2+1 | 0.1 | $63.2_{\pm 7.7}$ | ✗ |
| **ASAPool** | 1.11s | 1+1 | 0.1 | $83.5_{\pm 2.5}$ | ✗ |
| **SAGPool** | 0.59s | 1+1 | 0.1 | $79.5_{\pm 9.6}$ | ✗ |
| **Rand-dense** | 0.41s | 2+1 | 0.1 | $91.7_{\pm 1.3}$ | ✓ |
| **Cmp-Graclus** | 8.08s | 2+1 | 0.1 | $91.9_{\pm 1.2}$ | ✓ |
| **Rand-sparse** | 0.47s | 2+1 | 0.1 | $62.8_{\pm 1.8}$ | ✗ |

Table 1: Classification results on EXPWL1.

Table 1 also shows that employing a pooling operator based on a normalized random cluster assignment matrix (Rand-dense) or the complement graph (Cmp-Graclus) gives a lower performance. First of all, this result disproves the argument that such operators are comparable to the regular ones [28]. Additionally, we notice that the reduction in performance is less significant for Rand-Dense and Cmp-Graclus than for Rand-sparse. This outcome is expected because, in terms of expressiveness, Rand-dense and Cmp-Graclus still satisfy the conditions of Th. 1. Nevertheless, their performance is still lower than their regular counterparts. The reason is that even if a badly-behaved `CON` function does not compromise the expressiveness of the pooling operator, the structure of the pooled graph is corrupted when utilizing a randomized **S** or a complementary graph. This, in return, reduces the effectiveness of the last GIN layer, which is essential to correctly classify the graphs in EXPWL1.

There are two remarks about the experimental evaluation. As discussed in Section 3.2, it is not possible to explicitly specify the pooling ratio in Graclus, ECPool, and $k$-MISPool. For $k$-MISPool, setting $k = 5$ gives a pooling ratio of approximately 0.09 on EXPWL1. However, for Graclus, Cmp-Graclus, and ECPool, the only option is to apply the pooling operator recursively until the desired pooling ratio of 0.1 is reached. Unfortunately, this approach is demanding, both in terms of computing time and memory usage. While in EXPWL1 it was possible to do this for Graclus and Cmp-Graclus, we encountered an out-of-memory error after a few epochs when running ECPool on an RTX A6000 with 48GB of VRAM. Thus, the results for ECPool are obtained with a recursion that gives a pooling ratio of approximately 0.2. While this simplifies the training in ECPool we argue that, due to its expressiveness, ECPool would have reached approximately 100% accuracy on EXPWL1 if implementing a more aggressive pooling was feasible.

The second remark is that in EXPWL1 when using too many MP layers, at least one node ends up containing enough information to accurately classify the graphs. This was demonstrated a model with 3 GIN layers followed by `global_max_pool`, which achieved an accuracy of $98.3_{\pm 0.6}$ (more details in Appendix C.1). Note that the baseline model in Tab. 1 with 3 GIN layers equipped with the more expressive `global_sum_pool` achieves a slightly higher accuracy of $99.3_{\pm 0.3}$. In contrast, a model with only 2 GIN layers and `global_max_pool` gives a significantly lower accuracy of $66.5_{\pm 1.8}$. Therefore, to ensure that the evaluation is meaningful, no more than 2 MP layers should precede the pooling operator. Since ASAPool and SAGPool implement an additional MP operation internally, we used only 1 GIN layer before them, rather than 2 as for the other pooling methods.

Finally, Fig. 3 shows the average accuracy and the average run-time obtained on several benchmark datasets by a GNN equipped with the different pooling methods (the detailed results are in Appendix C.5). These benchmarks are not designed to test the expressive power and, thus, a GNN equipped with a non-expressive pooling operator could achieve good performance. Specifically, this happens in those datasets where all the necessary information is captured by the first two GIN layers that come before pooling or in datasets where only a small part of the graph is needed to determine the class. Nevertheless, this second experiment serves two purposes. First, it demonstrates the soundness of the GNN architecture used in the first experiment, which achieves results comparable to those of GNNs carefully optimized on the benchmark datasets [16]. Second, and most importantly, it shows that the performances on the benchmark datasets and EXPWL1 are aligned; this underlines the

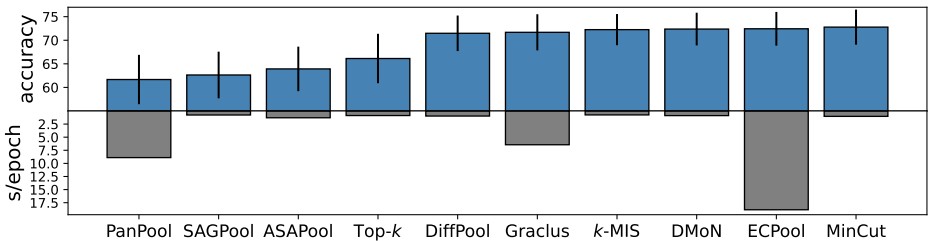

Figure 3: Average accuracy (and std.) v.s. average runtime on the benchmark datasets.

relevance of our theoretical result on the expressiveness in practical applications. It is worth noting that on the benchmark datasets, it was not possible to obtain a pooling ratio of 0.1 for both Graclus and ECPool. Using a pooling ratio of 0.5 gives Graclus and ECPool an advantage over other methods, which makes the comparison not completely fair and shows an important limitation of these two methods.

As a concluding remark, we comment on the training time of the dense and sparse pooling methods. A popular argument in favor of sparse pooling methods is their computational advantage compared to the dense ones. Our results show that this is not the case in modern deep-learning pipelines. In fact, ECPool, Graclus, PanPool, and even ASAPool are slower than dense pooling methods, while the only sparse method with training times lower than the dense ones is $k$-MIS. Even if it is true that the sparse methods save memory by avoiding computing intermediate dense matrices, this is relevant only for very large graphs that are rarely encountered in most applications.

## 5    Conclusions

In this work, we studied for the first time the expressive power of pooling operators in GNNs. We identified the sufficient conditions that a pooling operator must satisfy to fully preserve the expressive power of the original GNN model. Based on our theoretical results, we proposed a principled approach to evaluate the expressive power of existing graph pooling operators by verifying whether they met the conditions for expressiveness.

To empirically test the expressive power of a GNN, we introduced a new dataset that allows verifying if a GNN architecture achieves the same discriminative power of the WL test. We used such a dataset to evaluate the expressiveness of a GNN equipped with different pooling operators and we found that the experimental results were consistent with our theoretical findings. We believe that this new dataset will be a valuable tool as it allows, with minimal effort, to empirically test the expressive power of any GNN architecture. In our experimental evaluation, we also considered popular benchmark datasets for graph classification and found that the expressive pooling operators achieved higher performance. This confirmed the relevance in practical applications of our principled criterion to select a pooling operator based on its expressiveness. Finally, we focused on the computational time of the pooling methods and found that most sparse pooling methods not only perform worse due to their weak expressive power but are often not faster than the more expressive pooling methods.

We hope that our work will provide novel insights into the relational deep learning community and help to debunk misconceptions about graph pooling. We conclude by pointing to three limitations of our work. Firstly, the conditions of Th. 1 are sufficient but not necessary, meaning that there could be a non-expressive pooling operator that preserves all the necessary information. A similar consideration holds for EXPWL1: methods failing to achieve 100% accuracy are non-expressive, but the opposite is not necessarily true. In fact, reaching 100% accuracy on EXPWL1 is a necessary condition for expressiveness, though not sufficient. Secondly, condition 1 is not guaranteed to hold for continuous node features, which is a theoretical limitation not necessarily relevant in practice. Finally, our investigation focused on the scenario where the MP operation before pooling is as powerful as the 1-WL test. Even if layers more powerful than 1-WL test are rarely used in practice, it would be interesting to extend our approach to investigate the effect of pooling in these powerful architectures.

**Acknowledgements**

This research was partially funded by the MUR PNRR project "THE - Tuscany Health Ecosystem". We gratefully acknowledge the support of Nvidia Corporation with the donation of the RTX A6000 GPUs used in this work. Finally, we thank Daniele Zambon, Caterina Graziani, and Antonio Longa for the useful discussions.

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

# Appendices

## A  Proof of Theorem 1

*Proof.* Let $\mathbf{S} \in \mathbb{R}^{N \times K}$ and $\mathbf{T} \in \mathbb{R}^{M \times K}$ be the matrices representing the cluster assignments generated by $\text{SEL}(\mathcal{G}_1^L)$ and $\text{SEL}(\mathcal{G}_2^L)$, respectively. When condition 2 holds, we have that the entries of matrices $\mathbf{S}$ and $\mathbf{T}$ satisfy $\sum_{j=1}^{k} s_{ij} = \lambda, \forall i = 1, \ldots, N$ and $\sum_{j=1}^{K} t_{ij} = \lambda, \forall i = 1, \ldots, M$.

If condition 3 holds, then the $j$-th row of $\mathbf{X}_P$ is $\mathbf{x}_{P_j} = \sum_{i=1}^{N} \mathbf{x}_i^L \cdot s_{ij}$. The same holds for the $j$-th row of $\mathbf{Y}_P$, which is $\mathbf{y}_{P_j} = \sum_{i=1}^{M} \mathbf{y}_i^L \cdot t_{ij}$. Suppose that there exists a rows' permutation $\pi : \{1, \ldots, K\} \to \{1, \ldots, K\}$ such that $\mathbf{x}_{P_j} = \mathbf{y}_{P_{\pi(j)}}\ \forall i = 1, \ldots, M$, that is:

$$\sum_{i=1}^{N} \mathbf{x}_i^L \cdot s_{ij} = \sum_{i=1}^{M} \mathbf{y}_i^L \cdot t_{i\pi(j)} \quad \forall j = 1, \ldots, K$$

which implies

$$\sum_{j=1}^{K} \sum_{i=1}^{N} \mathbf{x}_i^L \cdot s_{ij} = \sum_{j=1}^{K} \sum_{i=1}^{M} \mathbf{y}_i^L \cdot t_{i\pi(j)} \Leftrightarrow \sum_{i=1}^{N} \mathbf{x}_i^L \cdot \sum_{j=1}^{K} s_{ij} = \sum_{i=1}^{M} \mathbf{y}_i^L \cdot \sum_{j=1}^{K} t_{i\pi(j)} \overset{2}{\Leftrightarrow}$$

$$\overset{2}{\Leftrightarrow} \sum_{i=1}^{N} \mathbf{x}_i^L \cdot \lambda = \sum_{i=1}^{M} \mathbf{y}_i^L \cdot \lambda \Leftrightarrow \sum_{i=1}^{N} \mathbf{x}_i^L = \sum_{i=1}^{M} \mathbf{y}_i^L$$

which contradicts condition 1. $\qquad\square$

Note that there are no restrictions on the cardinality of the original sets of nodes, $|\mathcal{V}_1| = N$ and $|\mathcal{V}_2| = M$. Indeed, since the proof does not depend on the number of nodes in the original graphs, $N$ can either be equal to $M$ or not. Additionally, we only focused on pooled graphs with the same number of nodes, i.e., $|\mathcal{V}_{1_P}| = |\mathcal{V}_{2_P}| = K$. The case where $|\mathcal{V}_{1_P}| \neq |\mathcal{V}_{2_P}|$ is trivial since two graphs with different numbers of nodes are inherently not WL equivalent.

## B  Examples from the EXPWL1 dataset

In Figure 4 we report four graph pairs from the EXPWL1 dataset. Each pair contains graphs with a different SAT outcome, which are WL-1 distinguishable. In the Figure, we used a different color map for each pair but the node features always assume a binary value in $\{0, 1\}$ in each graph.

## C  Experimental details

### C.1  Baseline models

To identify a GNN architecture that achieves approximately 100% accuracy on EXPWL1, we tried configurations with a different number of GIN layers followed by a `global_max_pool` or `global_sum_pool`. For the sake of comparison, we also considered a GNN with GCN layers, which are not expressive. The results are shown in Table 2.

As expected, the architectures with GIN layers outperform those with GCN layers, especially when the layers are two. This is due to the fact that GCN implements mean pooling as an aggregator, which is a well-defined multiset function due to its permutation invariance, but it lacks injectiveness, leading to a loss of expressiveness. Similarly, the GNNs with `global_sum_pool` perform better than those with `global_max_pool`, since the former is more expressive than the latter. An architecture with 3 GIN layers followed by `global_sum_pool` achieves approximately 100% accuracy on EXPWL1, making it the ideal baseline for our experimental evaluation. Perhaps more importantly, there is a significant difference in the performance of a 2-layers GNN followed by a global pooling layer that is more or less expressive. For this reason, the node embeddings generated by 2 GIN layers are a good candidate to test the expressiveness of the pooling operators considered in our analysis.

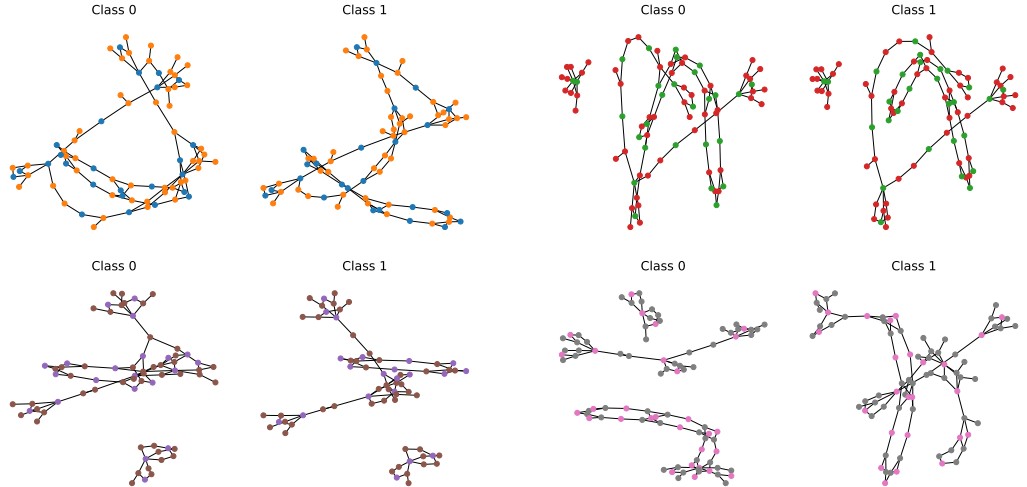

Figure 4: Four pairs of graphs from the EXPWL1 dataset. Each pair consists of two graphs with different classes that are 1-WL distinguishable.

| MP layers | Global Pool | Test Acc |
|-----------|-------------|----------|
| 2 GIN | global_max_pool | $66.5_{\pm 1.8}$ |
| 2 GIN | global_sum_pool | $92.1_{\pm 1.0}$ |
| 2 GCN | global_max_pool | $62.3_{\pm 2.4}$ |
| 2 GCN | global_sum_pool | $76.7_{\pm 2.4}$ |
| 3 GIN | global_max_pool | $98.3_{\pm 0.6}$ |
| 3 GIN | global_sum_pool | $99.3_{\pm 0.3}$ |
| 3 GCN | global_max_pool | $97.4_{\pm 0.5}$ |
| 3 GCN | global_sum_pool | $98.7_{\pm 0.6}$ |

Table 2: Performance of baseline architectures on EXPWL1.

## C.2 Hyperparameters of the GNN architectures

The GNN architecture used in all experiments consists of: [2 GIN layers] – [1 pooling layer with pooling ratio 0.1] – [1 GIN layer] – [global_sum_pool ] – [dense readout]. Each GIN layer is configured with an MLP with 2 hidden layers of 64 units and ELU activation functions. The readout is a 3-layer MLP with units [64, 64, 32], ELU activations, and dropout 0.5. The GNN is trained with Adam optimizer with an initial learning rate of 1e-4 using batches with size 32. For SAGPool or ASAPool we used only one GIN layer before pooling. For PanPool we used 2 PanConv layers with filter size 2 instead of the first 2 GIN layers. The auxiliary losses in DiffPool, MinCutPool, and DMoN are added to the cross-entropy loss with weights [0.1,0.1], [0.5, 1.0], [0.3, 0.3, 0.3], respectively. For $k$-MIS we used $k = 5$ and we aggregated the features with the sum. For Graclus, we aggregated the node features with the sum.

## C.3 EXPWL1 with different pooling ratios

In Table 3 we report the classification results obtained with different pooling ratios. Since in $k$-MIS we cannot specify the pooling ratio directly, we report the results obtained for $k = 3, 5, 6$ that gives approximately a pooling ratio of $0.19$, $0.09$, and $0.07$ respectively. As we discussed in the experimental section, Graclus, Cmp-Graclus, and ECPool roughly reduce the graph size by half. Therefore, to obtain the desired pooling ratio we apply them recursively. This creates a conspicuous computational overhead, especially in the case of ECPool which triggers an out-of-memory error for lower pooling ratios. Importantly, the results do not change significantly for the expressive pooling methods, while we notice a drastic improvement in the performance of the non-expressive ones for higher pooling ratios. Such sensitivity to different pooling ratios further highlights the practical difference between expressive and non-expressive operators.

| Pooling | Pool Ratio **0.05** | Pool Ratio **0.1** | Pool Ratio **0.2** |
|---|---|---|---|
| **DiffPool** | $95.2_{\pm2.1}$ | $97.0_{\pm2.4}$ | $97.4_{\pm2.2}$ |
| **DMoN** | $98.5_{\pm1.1}$ | $99.0_{\pm0.7}$ | $98.1_{\pm1.7}$ |
| **MinCut** | $98.3_{\pm0.7}$ | $98.8_{\pm0.4}$ | $98.4_{\pm0.6}$ |
| **ECPool** | OOM | OOM | $100.0_{\pm0.0}$ |
| **Graclus** | $100.0_{\pm0.0}$ | $99.9_{\pm0.1}$ | $99.9_{\pm0.1}$ |
| **$k$-MIS** | $99.8_{\pm0.2}$ | $99.9_{\pm0.1}$ | $100.0_{\pm0.0}$ |
| **Top-$k$** | $69.7_{\pm7.1}$ | $67.9_{\pm13.9}$ | $89.4_{\pm10.5}$ |
| **PanPool** | $61.3_{\pm5.5}$ | $63.2_{\pm7.7}$ | $75.4_{\pm12.8}$ |
| **ASAPool** | $84.3_{\pm2.5}$ | $83.5_{\pm2.5}$ | $87.3_{\pm7.2}$ |
| **SAGPool** | $61.6_{\pm10.6}$ | $79.5_{\pm9.6}$ | $82.4_{\pm11.1}$ |
| **Rand-dense** | $91.3_{\pm1.9}$ | $91.7_{\pm1.3}$ | $91.6_{\pm0.8}$ |
| **Cmp-Graclus** | $91.7_{\pm1.1}$ | $91.9_{\pm1.2}$ | $92.1_{\pm1.4}$ |
| **Rand-sparse** | $59.6_{\pm3.3}$ | $62.8_{\pm1.8}$ | $67.1_{\pm2.3}$ |

Table 3: Classification results on EXPWL1 using different pooling ratios.

## C.4 Statistics of the datasets

Table 4 reports the information about the datasets used in the experimental evaluation. Since the COLLAB and REDDIT-BINARY datasets lack vertex features, we assigned a constant feature value of 1 to all vertices.

| Dataset | #Samples | #Classes | Avg. #vertices | Avg. #edges | Vertex attr. | Vertex labels |
|---|---|---|---|---|---|---|
| **EXPWL1** | 3,000 | 2 | 76.96 | 186.46 | – | yes |
| **NCI1** | 4,110 | 2 | 29.87 | 64.60 | – | yes |
| **Proteins** | 1,113 | 2 | 39.06 | 72.82 | 1 | yes |
| **COLORS-3** | 10,500 | 11 | 61.31 | 91.03 | 4 | no |
| **Mutagenicity** | 4,337 | 2 | 30.32 | 61.54 | – | yes |
| **COLLAB** | 5,000 | 3 | 74.49 | 4,914.43 | – | no |
| **REDDIT-B** | 2,000 | 2 | 429.63 | 995.51 | – | no |
| **B-hard** | 1,800 | 3 | 148.32 | 572.32 | – | yes |
| **MUTAG** | 188 | 2 | 17.93 | 19.79 | – | yes |
| **PTC_MR** | 344 | 2 | 14.29 | 14.69 | – | yes |
| **IMDB-B** | 1,000 | 2 | 19.77 | 96.53 | – | no |
| **IMDB-MULTI** | 1,500 | 3 | 13.00 | 65.94 | – | no |
| **ENZYMES** | 600 | 6 | 32.63 | 62.14 | 18 | yes |
| **REDDIT-5K** | 4,999 | 5 | 508.52 | 594.87 | – | no |

Table 4: Details of the graph classification datasets.

## C.5 Detailed performance on the benchmark datasets

| DiffPool | DMoN | MinCut | ECPool | Graclus | $k$-MIS | Top-$k$ | PanPool | ASAPool | SAGPool |
|---|---|---|---|---|---|---|---|---|---|
| 0.96s | 0.88s | 1.02s | 18.85s | 6.44s | 0.75s | 0.87s | 8.89s | 1.29s | 0.76s |
| $71.4_{\pm3.7}$ | $72.3_{\pm3.4}$ | $72.8_{\pm3.7}$ | $72.4_{\pm3.5}$ | $71.6_{\pm3.8}$ | $72.2_{\pm3.3}$ | $66.1_{\pm5.2}$ | $61.6_{\pm5.2}$ | $63.9_{\pm4.7}$ | $62.6_{\pm4.9}$ |

Table 5: Average run-time in seconds per epoch (first row) and average classification accuracy (second row) achieved by the different pooling methods on the benchmark datasets.

The test accuracy of the GNNs configured with the different pooling operators on the graph classification benchmarks is reported in Table 6, while Table 7 reports the run-time of each model expressed in seconds per epoch. The overall average accuracy and average run-time computed across all datasets are presented in Table 5. For each dataset, we use the same GNN configured as described in C.2, as we are interested in validating the architecture used to classify EXPWL1. Clearly, by using less aggressive pooling, by fine-tuning the GNN models, and by increasing their capacity it is possible to improve the results on several datasets. Such results are reported in the original papers introducing the different pooling operators.

| Pooling | NCI1 | PROTEINS | COLORS-3 | Mutagenity | COLLAB | REDDIT-B | B-hard |
|---|---|---|---|---|---|---|---|
| **DiffPool** | $77.8_{\pm3.9}$ | $72.8_{\pm3.3}$ | $87.6_{\pm1.0}$ | $80.0_{\pm1.9}$ | $76.6_{\pm2.5}$ | $89.9_{\pm2.8}$ | $70.2_{\pm1.5}$ |
| **DMoN** | $78.5_{\pm1.4}$ | $73.1_{\pm4.6}$ | $88.4_{\pm1.4}$ | $81.3_{\pm0.3}$ | $80.9_{\pm0.7}$ | $91.3_{\pm1.4}$ | $71.1_{\pm1.0}$ |
| **MinCut** | $80.1_{\pm2.6}$ | $76.0_{\pm3.6}$ | $88.7_{\pm1.6}$ | $81.2_{\pm1.9}$ | $79.2_{\pm1.5}$ | $91.9_{\pm1.8}$ | $71.2_{\pm1.1}$ |
| **ECPool** | $79.8_{\pm3.3}$ | $69.5_{\pm5.9}$ | $81.4_{\pm3.3}$ | $82.0_{\pm1.6}$ | $80.9_{\pm1.4}$ | $90.7_{\pm1.7}$ | $74.5_{\pm1.6}$ |
| **Graclus** | $81.2_{\pm3.4}$ | $73.0_{\pm5.9}$ | $77.6_{\pm1.2}$ | $81.9_{\pm1.6}$ | $80.4_{\pm1.5}$ | $92.9_{\pm1.7}$ | $72.3_{\pm1.3}$ |
| $k$-**MIS** | $77.6_{\pm3.0}$ | $75.9_{\pm2.9}$ | $82.9_{\pm1.7}$ | $82.6_{\pm1.2}$ | $73.7_{\pm1.4}$ | $90.6_{\pm1.4}$ | $71.7_{\pm0.9}$ |
| **Top-**$k$ | $72.6_{\pm3.1}$ | $73.2_{\pm2.7}$ | $57.4_{\pm2.5}$ | $74.4_{\pm4.7}$ | $77.9_{\pm2.1}$ | $87.4_{\pm3.5}$ | $68.1_{\pm7.7}$ |
| **PanPool** | $66.1_{\pm2.3}$ | $75.2_{\pm6.2}$ | $40.7_{\pm11.5}$ | $67.2_{\pm2.0}$ | $78.2_{\pm1.5}$ | $83.6_{\pm1.9}$ | $44.2_{\pm8.5}$ |
| **ASAPool** | $73.1_{\pm2.5}$ | $75.5_{\pm3.2}$ | $43.0_{\pm4.7}$ | $76.5_{\pm2.8}$ | $78.4_{\pm1.6}$ | $88.0_{\pm5.6}$ | $67.5_{\pm6.1}$ |
| **SAGPool** | $79.1_{\pm3.0}$ | $75.2_{\pm2.7}$ | $43.1_{\pm11.1}$ | $77.9_{\pm2.8}$ | $78.1_{\pm1.8}$ | $84.5_{\pm4.4}$ | $54.0_{\pm6.6}$ |
| **Rand-dense** | $78.2_{\pm2.0}$ | $75.3_{\pm1.3}$ | $83.3_{\pm0.9}$ | $81.4_{\pm1.8}$ | $69.3_{\pm1.6}$ | $89.3_{\pm2.1}$ | $71.0_{\pm2.2}$ |
| **Cmp-Graclus** | $77.8_{\pm1.8}$ | $73.6_{\pm4.7}$ | $84.7_{\pm0.9}$ | $80.7_{\pm1.8}$ | OOM | OOM | OOM |
| **Rand-sparse** | $69.1_{\pm3.3}$ | $74.6_{\pm4.2}$ | $35.5_{\pm1.1}$ | $69.8_{\pm1.0}$ | $68.8_{\pm1.6}$ | $84.5_{\pm1.9}$ | $50.1_{\pm4.0}$ |

| Pooling | MUTAG | PTC_MR | IMDB-B | IMDB-MULTI | ENZYMES | REDDIT-5K |
|---|---|---|---|---|---|---|
| **DiffPool** | $86.8_{\pm9.7}$ | $54.7_{\pm6.1}$ | $71.3_{\pm3.1}$ | $45.2_{\pm3.4}$ | $62.3_{\pm7.3}$ | $53.7_{\pm1.8}$ |
| **DMoN** | $86.3_{\pm7.1}$ | $55.5_{\pm7.3}$ | $71.9_{\pm3.3}$ | $47.0_{\pm5.5}$ | $61.0_{\pm5.0}$ | $56.6_{\pm2.3}$ |
| **MinCut** | $83.1_{\pm9.6}$ | $57.9_{\pm7.7}$ | $71.9_{\pm5.7}$ | $46.6_{\pm4.0}$ | $62.3_{\pm3.8}$ | $56.2_{\pm2.8}$ |
| **ECPool** | $90.0_{\pm7.2}$ | $54.7_{\pm8.4}$ | $71.3_{\pm3.4}$ | $49.2_{\pm2.9}$ | $59.6_{\pm3.7}$ | $53.6_{\pm2.2}$ |
| **Graclus** | $85.2_{\pm8.0}$ | $55.2_{\pm6.4}$ | $72.3_{\pm5.8}$ | $46.2_{\pm4.4}$ | $61.0_{\pm6.6}$ | $52.3_{\pm1.4}$ |
| $k$-**MIS** | $85.7_{\pm6.2}$ | $59.7_{\pm5.7}$ | $73.1_{\pm4.2}$ | $46.8_{\pm4.6}$ | $63.5_{\pm7.1}$ | $56.4_{\pm2.3}$ |
| **Top-**$k$ | $78.4_{\pm11.8}$ | $58.2_{\pm8.9}$ | $70.9_{\pm3.3}$ | $44.8_{\pm2.9}$ | $45.5_{\pm10.5}$ | $50.4_{\pm3.7}$ |
| **PanPool** | $83.1_{\pm13.2}$ | $53.5_{\pm7.7}$ | $73.9_{\pm3.5}$ | $48.3_{\pm3.7}$ | $40.5_{\pm5.0}$ | $46.5_{\pm2.4}$ |
| **ASAPool** | $74.2_{\pm6.8}$ | $50.5_{\pm12.1}$ | $71.4_{\pm2.8}$ | $46.1_{\pm4.2}$ | $44.8_{\pm7.6}$ | $48.8_{\pm1.6}$ |
| **SAGPool** | $73.7_{\pm6.6}$ | $58.8_{\pm8.0}$ | $71.0_{\pm4.0}$ | $44.0_{\pm3.4}$ | $41.6_{\pm5.2}$ | $49.9_{\pm2.9}$ |
| **Rand-dense** | $88.9_{\pm4.3}$ | $56.1_{\pm9.7}$ | $70.5_{\pm3.4}$ | $45.2_{\pm5.6}$ | $62.1_{\pm5.0}$ | $54.5_{\pm2.1}$ |
| **Cmp-Graclus** | $83.2_{\pm9.1}$ | $55.9_{\pm4.6}$ | OOM | OOM | $63.5_{\pm5.0}$ | OOM |
| **Rand-sparse** | $68.9_{\pm17.3}$ | $56.4_{\pm5.9}$ | $71.6_{\pm3.6}$ | $45.8_{\pm3.7}$ | $62.1_{\pm5.0}$ | $50.6_{\pm2.4}$ |

Table 6: Graph classification test accuracy on benchmark datasets.

| Pooling | NCI1 | PROTEINS | COLORS-3 | Mutagenity | COLLAB | REDDIT-B | B-hard |
|---|---|---|---|---|---|---|---|
| **DiffPool** | 0.83s | 0.23s | 1.67s | 0.90s | 1.68s | 1.74s | 0.29s |
| **DMoN** | 1.01s | 0.28s | 1.94s | 1.06s | 1.83s | 1.04s | 0.33s |
| **MinCut** | 0.95s | 0.28s | 1.82s | 1.10s | 1.82s | 1.78s | 0.35s |
| **ECPool** | 4.39s | 1.97s | 10.30s | 4.22s | 44.11s | 3.17s | 6.90s |
| **Graclus** | 0.95s | 0.27s | 2.47s | 0.98s | 3.01s | 0.75s | 0.31s |
| $k$-**MISPool** | 0.88s | 0.25s | 2.48s | 0.95s | 1.38s | 0.48s | 0.43s |
| **Top-**$k$ | 1.04s | 0.29s | 2.78s | 1.04s | 2.79s | 0.47s | 0.30s |
| **PanPool** | 2.81s | 0.81s | 7.16s | 5.48s | 7.67s | 46.15s | 6.27s |
| **ASAPool** | 1.83s | 0.52s | 4.48s | 1.80s | 3.97s | 0.79s | 0.52s |
| **SAGPool** | 1.09s | 0.30s | 2.52s | 1.07s | 2.81s | 0.43s | 0.28s |
| **Rand-dense** | 0.54s | 0.14s | 1.44s | 0.55s | 0.88s | 1.44s | 0.26s |
| **Cmp-Graclus** | 7.94s | 3.27s | 34.05s | 1.94s | – | – | – |
| **Rand-sparse** | 0.64s | 0.18s | 1.72s | 0.68s | 1.00s | 0.47s | 0.31s |

| Pooling | MUTAG | PTC_MR | IMDB-B | IMDB-MULTI | ENZYMES | REDDIT-5K |
|---|---|---|---|---|---|---|
| **DiffPool** | 0.04s | 0.07s | 0.14s | 0.28s | 0.12s | 4.52s |
| **DMoN** | 0.05s | 0.09s | 0.17s | 0.37s | 0.15s | 3.21s |
| **MinCut** | 0.04s | 0.08s | 0.16s | 0.35s | 0.14s | 4.45s |
| **ECPool** | 0.08s | 0.12s | 1.66s | 1.01s | 0.54s | 37.13s |
| **Graclus** | 0.09s | 0.14s | 0.24s | 0.33s | 0.14s | 74.02s |
| $k$-**MIS** | 0.06s | 0.10s | 0.28s | 0.21s | 0.12s | 2.02s |
| **Top-**$k$ | 0.04s | 0.08s | 0.24s | 0.32s | 0.13s | 1.75s |
| **PanPool** | 0.26s | 0.49s | 1.30s | 1.92s | 0.98s | 34.3s |
| **ASAPool** | 0.14s | 0.10s | 0.43s | 0.40s | 0.16s | 1.89s |
| **SAGPool** | 0.04s | 0.05s | 0.23s | 0.27s | 0.08s | 1.09s |
| **Rand-dense** | 0.04s | 0.06s | 0.13s | 0.19s | 0.12s | 3.48s |
| **Cmp-Graclus** | 0.17s | 0.47s | – | – | 1.38s | – |
| **Rand-sparse** | 0.04s | 0.08s | 0.15s | 0.23 | 0.14s | 1.67s |

Table 7: Graph classification test run-time in s/epoch.

