# OpenReview forum: "The expressive power of pooling in Graph Neural Networks"
_NeurIPS.cc/2023/Conference — NeurIPS 2023 poster_

### Official Review · Reviewer_QuJp · 2023-07-01

**Soundness:** 3 good
**Presentation:** 3 good
**Contribution:** 3 good
**Rating:** 6
**Confidence:** 3

**Summary:**

The paper analyzes the expressive power of pooling (not to be confused with readout) in message passing GNNs (MP-GNNs). The paper gives a condition (in Theorem 1) on the construction of the POOL function, under which there is a choice of the MP-GNN which can separate the same graphs that the 1-WL test can separate. Using these conditions, well known pooling methods are classified as expressive and non-expressive. A new dataset of pairs of graphs that can be separated by 1-WL is introduced, on which the theoretical results are demonstrated.

**Strengths:**

The paper is mostly clear, and the topic of pooling in GNNs has practical importance. The theoretical analysis is sound (up to some issue with its interpretation that I discuss below). The classification of well known pooling methods as expressive and non-expressive, in view of Theorem 1, is potentially useful for practitioners, especially as the experimental results on the new dataset strongly support this theoretical classification. The approach is a natural extension of expressivity analysis such as in "How powerful are graph neural networks?" and “Deep sets.”


**Weaknesses:**

The discussion on the injectivity of pooling should be put in the right context and setting. What is shown around Theorem 1 is that there is a specific way to predefine the MP functions in such a way that the whole network, including POOL, is injective. The construction does not show that POOL is injective for general message passing functions with trainable parameters (for any choice of the parameters). See more details on this point in “Questions” below. Once this is fixed, I can improve my rating.

**Questions:**

I would replace the non-standard notations of multi-sets and multi-sets of multi-sets with standard notations of 1D and 2D arrays, as these are the objects you actually deal with in this paper. For example:

Line 65: the node features should be represented as a vector (ordered sequence) of feature values, which is equivalent to a mapping from the nodes to features, not as a multiset. A multiset does not have the information of which feature belongs to which node.  Same in line 99, and throughout the paper. For example, 3 in line 130 is a 2D array, not a multiset of multisets, since nodes i and j have identities.

Theorem 1 is important and correct, but it cannot directly be called an expressivity theorem.
The logic in the paragraph of line 133 is backwards in a sense. In condition 1 of Theorem 1 you ASSUME a condition on the input of pooling that is similar, or at least stronger, than “two feature vectors on which pooling gives two different outputs.’’ It is almost like giving a condition for the injectivity of a function f by saying “the function f(x) gives different values on points such that $f(x)\neq f(y)$.’’ Injectivity is not a property of specific inputs, it is a property of the function. An appropriate injectivity analysis should say that, under some assumptions on the model of pooling, for ANY two distinct graphs+node features, the pooling of the two gives two distinct graphs+node features. The theorem you stated is a theorem about which inputs can be separated by pooling, not about which pooling method can separate any pair of inputs.
The theorem is still meaningful, as it is used later to discuss expressivity in case features are computed in such a way that sum is injective, but it should be described in the correct context.
In this context: in Line 135:
“there are theorems for functions defined on sets that guarantee condition 1 to be met. In particular, the sum over a multiset that is countable is an injective function”
There are indeed very specific (using very high dimensional feature spaces) ways to design by hand specific MP mappings that satisfy this, but when training a MP-GNN there is no known (at least to my knowledge) practical way to guarantee this property.
Hence, this paragraph should also be put in the right context. Namely, there is a way to predefine the MP functions such that POOL is injective (when combined with these predefined MP functions). This is different from saying that POOL is injective on a trained network, or a network with generic weights.
To make this clear, it would be good to write an appendix that explains in detail the description around line 133, that is currently quite implicit, citing results from [37] and [40] and formulating the results in a mathematical language. The appendix should show that there exists a construction of MP-GNN with pooling which separates points. This is a type of analysis that directly extends the expressivity analysis of [37] to GNNs with pooling.

In the new dataset, what feature is put on the nodes of the graphs?

Table 1 - can you add for comparison non expressive networks like GCN (without pooling), so we see what the “baseline” is, and how hard it is to separate graphs when you do not use pooling?


Minor comments and suggestions:

Line 129 : (suggested terminology) this is called a right stochastic matrix up to the global constant $\lambda$.

Line 3: this is the multiplication of the feature vector from the left by the right stochastic matrix S.



**Limitations:**

The paper should clarify that expressivity does not mean that POOL is injective for some class of MP function, but that there is a specific choice of the parameters of the MP function such that POOL is injective, similarly to the analysis of [37] and [40].

---

> ### Author Rebuttal · Authors · 2023-08-05
>
> Thank you for the thoughtful comments and detailed review.
>
> > Put the injectivity of pooling in the right context and setting. The construction does not show that POOL is injective for general message-passing functions with trainable parameters.
>
> We completely agree that the objective of Theorem 1 is not to establish sufficient conditions for the injectivity of the pooling operator.
> Indeed, our goal is to study the expressive capabilities of pooling layers **within** a GNN architecture. Our focus lies in identifying sufficient conditions under which a set of MP parameters exists, leading to an injective composition of message-passing and pooling layers.
>
> We have provided further clarification on this matter in Section 3.1.
>
> > Replace the notation of multi-sets and multi-sets of multi-sets with standard notations of 1D and 2D arrays.
>
> We agree that 1D and 2D arrays are what we deal with.
> However, since the GNNs that we consider implement operations that are permutation invariant, node features can be equivalently represented with multisets.
> We found it advantageous since the multisets notation facilitates both the discussion and the formal derivations.
> We clarified this in section 2.1 where we introduce the notation.
>
> > Injectivity is a property of the function, not of the input.
>
> We agree that injectivity is a property of the function, and not of the input. However, we would like to clarify that the function for which we aim to ensure injectivity is not just the POOL function, but is the **composition of MP layers and pooling layers**. We would like to stress that none of the three conditions put constraints on the input of the architecture composed of MP layers and pooling layers. In particular, the input of MP layers and pooling layer can be any graph. Then, the three conditions guarantee that for any two distinct graphs+node features, the composition of MP layers and pooling gives two distinct graphs+node features.
>
> > Put in the right context Theorem 1 and the discussion that follows.
>
> These comments were very useful to us. Thank you.
>
> We clarified that Theorem 1 ensures that there exists a specific choice of parameters for the MP layer that, when combined with a pooling operator satisfying the Theorem's conditions, the resulting GNN architecture is injective. In particular, we modified the first paragraph of the discussion about Th.1 as follows.
>
> *Condition 1 is strictly related to the theory of multisets. Indeed, a major breakthrough in designing highly expressive MP layers was achieved by building upon the findings of Deep Sets [40].
> Under the assumption that the node features originate from a countable universe, it has been formally proven that there exists a function that, when applied to the node features, makes the sum over a multiset of node features injective [37]. The universal approximation theorem guarantees that this function can be modeled using an MLP. Moreover, if the pooling operator satisfies conditions 2 and 3, it will produce multisets of node features so that $\mathcal{X}_ {1_P} \neq \mathcal{X}_ {2_P}$.  Due to the injectiveness of the coloring function of the WL algorithm, two graphs with different multisets of node features will be classified as non-isomorphic by the WL test and, therefore, $\mathcal{G}_ {1_P} \neq_\text{WL} \mathcal{G}_ {2_P}$. This implies that the pooling operator effectively coarsens the graphs while retaining all the information necessary to distinguish them.  Therefore, our Theorem ensures that there exists a specific choice of parameters for the MP layer that, when combined with a pooling operator satisfying the Theorem's conditions, the resulting GNN architecture is injective.*
>
> We also added more details from references [37] and [40] in the appendix.
>
> [37] Xu et al., How powerful are graph neural networks?, 2019
>
> [40] Zaheer et al., Deep Sets, 2017
>
> > Node features in the new dataset
>
> The node features assume a binary value in {0,1} in each graph. We clarified it and showed examples of graph pairs from different classes. See Figure 4 in the PDF attached as part of the rebuttal.
>
> > Baseline with GCN layers.
>
> We performed the experiment requested. The results are in Table 2 in the PDF attached.
>
> > Minor comments and suggestions.
>
> We implemented them, thanks.

---

> > ### Comment · Reviewer_QuJp · 2023-08-10
> > **Issue with modeling the features as a multiset**
> >
> > Thank you for the clarifications and revisions in the paper. All issues were addressed apart from one.
> >
> > The issue with multisets still remains. You can indeed model the features *within each neighborhood* as a multiset, but you cannot model the features of the whole graph as a multiset. Each feature must be assigned to a node. Take for example a grid as the graph, where the node features represent a gray level image. Suppose we want to represent an image of a white disc on black background, where there are a total of 100 pixels. Suppose that half the pixels are inside the disc and half are outside. If you try to represent this signal as a multiset, you would just get a multiset with 50 feature of values 1 and 50 features of value 0. But nothing tells you to which node each feature belongs. An image that looks like a 010101010 patten in both axes has the exact same multiset of features, but the signal is very different. A GNN can distinguish between these two images easily, so modeling the signal as a multiset is not appropriate.
> >
> > You need to change the data-structure of the features from a multiset to a vector for the mathematical formulation to be rigorous.

---

> > > ### Author Response · Authors · 2023-08-11
> > >
> > > We are really grateful to you for having taken the time to carefully read our paper, the rebuttal, and to explain your points.
> > > You are right, it is not correct to use multisets to express the input and the output of a GNN as it implements equivariant operations.
> > >
> > > We have removed everywhere in the paper the multiset notation. After changing the notation, we slightly modified the formulation of condition 3 and the Theorem's conclusion, which we report below for completeness.
> > > The only significant change is that when using matrix notation rather than multiset notation, we had to specify that the two matrices $\mathbf{X}_ {P}$ and $\mathbf{Y}_ {P}$ remain different no matter how their rows are permuted.
> > > The other adjustments due to the change in notation, including those of the proof, are rather straightforward.
> > > Thanks again for your valuable feedback.
> > >
> > > ---
> > > **Theorem 1 (with new notation):**
> > >
> > > *Let $\mathcal{G}_ 1=(\mathcal{V}_ 1, \mathcal{E}_ 1)$ with $|\mathcal{V}_ 1|=N$ and $\mathcal{G}_ 2=(\mathcal{V}_ 2, \mathcal{E}_ 2)$ with $|\mathcal{V}_ 2|=M$ with node features $\mathbf{X}$ and $\mathbf{Y}$ respectively,  such that $\mathcal{G}_ 1\neq_{WL}\mathcal{G}_ 2$.
> > > Let $\mathcal{G}^L_ 1$ and $\mathcal{G}^L_ 2$ be the graph obtained after applying a block of $L$ MP layers such that $\mathbf{X}^L\in \mathbb{R}^{N\times F}$ and $\mathbf{Y}^L \in \mathbb{R}^{M\times F} $ are the new nodes features. Let $\texttt{POOL}$ be a pooling operator expressed by the functions $\texttt{SEL}, \texttt{RED}, \texttt{CON}$, which is placed after the MP layers. Let $\mathcal{G}_ {1_P}=\texttt{POOL}(\mathcal{G}_ 1)$ and  $\mathcal{G}_ {2_P}=\texttt{POOL}(\mathcal{G}_ 2)$ with $|\mathcal{V}_ {1_P}|=|\mathcal{V}_ {2_P}|=K$.
> > > Let $\mathbf{X}_ {P}\in \mathbb{R}^{K\times F}$ and $\mathbf{Y}_ {P}\in \mathbb{R}^{K\times F}$ be the nodes features of the pooled graphs, i.e, $\mathbf{x}_ {P_j}$ is the features of the supernode $j$ in graph $\mathcal{G}_ {1_P}$, and $\mathbf{y}_ {P_j}$ is the features of the supernode $j$ in graph $\mathcal{G}_ {2_P}$. If the following conditions hold:*
> > >
> > > 1. *$\sum_i^N \mathbf{x}_i^L \neq \sum_i^M \mathbf{y}^L_i $;*
> > > 2. *For each node $i$, the memberships generated by $\texttt{SEL}$ satisfy $\sum_{j=1}^{K} s_{ij}=\lambda$, with $\lambda>0$, i.e., the cluster assignment matrix \textbf{S} is a right stochastic matrix up to the global constant $\lambda$;*
> > > 3. *The function $\texttt{RED}$ is of type $\texttt{RED}: (\mathbf{X}^L,\mathbf{S})\mapsto \mathbf{X}_P=\mathbf{S}^T \mathbf{X}^L$;*
> > >
> > > *then $\mathcal{G}_ {1_P}$ and $\mathcal{G}_ {2_P}$ will have different nodes features, i.e., for all rows' indices permutations $\pi: \{1, \ldots K\} \rightarrow \{1, \ldots K\}$,
> > > $\mathbf{X}_ {P}\neq \Pi(\mathbf{Y}_ {P})$, where $[\Pi(\mathbf{Y}_ {P})]_ {ij} = \mathbf{y}_ {P_ {\pi(i)j}}$.*

---

> > > > ### Comment · Reviewer_QuJp · 2023-08-11
> > > >
> > > > Thank you for the response. I increase my rating to 6, conditioned on the changes being implemented in the camera ready paper.

---

### Official Review · Reviewer_UT1a · 2023-07-04

**Soundness:** 4 excellent
**Presentation:** 4 excellent
**Contribution:** 3 good
**Rating:** 7
**Confidence:** 4

**Summary:**

The authors present a comprehensive analysis of pooling operations in Graph Neural Networks (GNNs) from both theoretical and practical perspectives. Additionally, they introduce a refined dataset, EXPWL1, specifically designed to facilitate the analysis of the expressiveness of pooling operations. Remarkably, the empirical evaluation aligns with the theoretical discoveries.

**Strengths:**

1. The content is presented in a clear and organized manner, and the figures are well-drawn and visually appealing.
2. The theoretical portion of the work is effectively presented and explained.
3. The experimental section contains a thorough explanation of the setup and a comprehensive evaluation of the results.


**Weaknesses:**

1. The discussion on the limitations of the work is absent.
2. Figure 3 is missing std bars.


**Questions:**


1. In line 128 of Theorem 1, it is unclear what the inequality refers to. Could you please clarify if each element of the vector needs to be different or if it is sufficient for only one of them to be different?

2. Regarding the limitations, I would like to inquire whether the results apply universally to cases where the features of the nodes are continuous.

3. In Figure 3, if my understanding is correct, each bar represents the average accuracy (in blue) and runtime (in gray) across six different datasets. It would be beneficial to include the standard deviation of these averages. Specifically, the accuracy values may exhibit significant variability between datasets.

MINOR POINTS

- There is a typo in Equation (2), with a missing opening curly bracket.

**Limitations:**

The authors have not explicitly mentioned the limitations of their work, particularly in terms of the cases where their theory does not apply and any potential limitations of the proposed EXPWL1 method. It would be beneficial for the authors to address these aspects in order to provide a comprehensive understanding of the scope and potential constraints of their research.

---

> ### Author Rebuttal · Authors · 2023-08-04
>
> Thanks for the positive evaluation of the paper and for the useful suggestions.
>
> > Discuss the limitations and whether the results apply to continuous features.
>
> Thanks for the suggestion. In the conclusions, we now discuss the following limitations:
>
> "*Firstly, the conditions of Th.1 are sufficient but not necessary, meaning that a non-expressive pooling operator could be able to preserve all the necessary information. A similar consideration holds for EXPWL1: methods failing to achieve 100% accuracy are non-expressive, but the opposite is not necessarily true. In fact, reaching 100% accuracy is a necessary condition for expressiveness, though not sufficient. Secondly, condition 1 is not guaranteed to hold for continuous node features, which is a theoretical limitation not necessarily relevant in practice. Finally, our investigation focused on the scenario where the MP operation before pooling is as powerful as the 1-WL test.  Even if layers more powerful than 1-WL test are rarely used in practice, it would be interesting to extend our approach to investigate the effect of pooling in these powerful architectures.*"
>
> > Clarify line 128 of Theorem 1.
>
> Thanks for the suggestion. We clarified that it is sufficient that only one element of the vectors is different.
>
> > Include standard deviations in Figure 3.
>
> Thanks for the suggestion. We included error bars and enlarged the fonts to make the figure more readable. The updated figure is in the PDF attached to the rebuttal.
>
> > Typo in Equation 2.
>
> Thanks, we corrected the typo.

---

### Official Review · Reviewer_KM5y · 2023-07-05

**Soundness:** 3 good
**Presentation:** 3 good
**Contribution:** 2 fair
**Rating:** 6
**Confidence:** 5

**Summary:**

This paper analyzes the expressive power of pooling operators in Graph Neural Networks (GNNs) and derives three conditions for a pooling operator to fully preserve the expressive power of the Message Passing (MP) layers preceding it. The derived conditions are as follows:

a) The pooling operator should extract different features for non-isomorphic graphs that are distinguishable using the Weisfeiler-Lehman (WL) test.

b) All nodes in the original graph must contribute to the creation of supernodes.

c) The features of the supernodes (denoted as X^P) should be a convex combination of the features of the nodes X^L.

The paper presents controlled experiments along with theoretical support to validate and support their findings.

**Strengths:**

1 - The paper is well-written and easy to follow.
2 - The hypothesis is nicely executed in a controlled set of experiments.
3 - Theoretical analysis has been provided to support the findings.
4 - Sufficient review of the literature and pooling operators is provided.
5 - Clear categorization and analysis of the pooling operators are provided.

**Weaknesses:**

1 - Selected real-world graph classification datasets are chosen. I was wondering why MUTAG, PTC_MR, IMDB-B are IMDB-MULTI left out
2 - A comparison with pooling operators such as global_add, global_sum and global_max might provide a better overview of the effectiveness of these pooling operators.

**Questions:**

1 - I found the figure 2 slightly difficult to understand. I believe that adding some additional information, if applicable, could be helpful in making it easier to comprehend.

2 - I appreciate that the running times are provided, but I was wondering if it would be possible to derive their computational complexities (pooling operators) and include them in Table 2.

3 - The quality of Figure 3 could potentially be improved.

**Limitations:**

-

---

> ### Author Rebuttal · Authors · 2023-08-04
>
> Thanks for your comments and suggestions.
>
> > MUTAG, PTC-MR, IMDB-B, and IMDB-MULTI were left out.
>
> We performed the experiments on these datasets. In addition, we also performed experiments on ENZYMES and REDDIT-5K (the latter is still running and will be completed in a few days). Table 5 with the updated results is in the PDF attached to the rebuttal.
>
> > Comparison of different global pooling operators.
>
> Please note that in the experimental section, we were already discussing the results obtained using global-sum-pool rather than global-max-pool.
> On top of that, now we added Table 2 (see attached PDF) which shows a detailed comparison between architectures with GIN e GCN layers (which were asked by another reviewer) followed by global-max-pool and global-sum-pool.
>
> > Figure 2 is difficult to understand.
>
> Thanks for the suggestion. We simplified the figure and we provided a more detailed description in the caption. The new figure is in the PDF attached.
>
> > Adding computational complexities in Table 2.
>
> We reviewed several papers to obtain the computational complexities of the pooling methods under consideration. Dense pooling methods typically have a computational complexity of $\mathcal{O}(k(m + nk))$ where $n$ is the number of nodes, $m$ is the number of edges and $k$ is the number of supernodes. In contrast, sparse pooling methods generally have a complexity $\mathcal{O}(m+n)$. However, each paper relies on different definitions and calculations to estimate the complexity, making a direct comparison challenging and prone to errors. For example, [1] and [2] report different complexities for MinCut, while [3] and [5]  report different complexities for ECPool. In addition, the complexity of some methods depends on specific software implementations and hardware acceleration. Indeed, the actual computing times do not reflect well the differences in the algorithmic complexity. For these reasons, we found it more meaningful and transparent to report the computing time obtained with the methods implemented in Pytorch Geometric.
>
> [1] Bianchi et al., Spectral clustering with graph neural networks for graph pooling, 2020.
>
> [2] Tsitsulin et al., Graph Clustering with Graph Neural Networks, 2023.
>
> [3] Landolfi, Revisiting Edge Pooling in Graph Neural Networks, 2022.
>
> [4] Dong at al., MeGraph: Graph Representation Learning on Connected Multi-scale Graphs, 2023.
>
> > Improve the quality of Figure 3.
>
> Thanks for the suggestion. We included error bars and enlarged the fonts to make Fig. 3 more readable. The updated figure is in the PDF attached.

---

> ### Comment · Reviewer_KM5y · 2023-08-13
> **Response to authors on rebuttal**
>
> The authors have done a great job in addressing my comments. I have no further comments and increasing my score to 6.

---

### Official Review · Reviewer_DheR · 2023-07-27

**Soundness:** 3 good
**Presentation:** 4 excellent
**Contribution:** 3 good
**Rating:** 6
**Confidence:** 4

**Summary:**

This paper presents a study on the performance and expressiveness of various pooling operators in Graph Neural Networks (GNNs) in both theoretical and empirical ways. In detail, the authors identify the sufficient conditions that a pooling operator must satisfy to fully preserve the expressive power of the original GNN model. The authors also propose an experimental approach to evaluate the performance of various graph pooling operators and their theoretical results. The empirical results align with the theoretical findings, showing that pooling operators that satisfy the aforementioned conditions achieve the highest average accuracy. Despite aggressive pooling, these operators retain all the necessary information and perform as well as the GNN without a pooling layer. On the other hand, non-expressive pooling operators achieve significantly lower accuracy. The paper also demonstrates that pooling operators based on a normalized random cluster assignment matrix or the complement graph yield lower performance, which challenges the notion that such operators are comparable to regular ones. In summary, this paper makes a valuable contribution to the field of GNNs by investigating the expressive power of pooling operators in GNNs.

**Strengths:**

**Originality**: The paper is highly original in its focus on the performance and expressiveness of various pooling operators in GNNs. The authors propose a novel experimental approach to evaluate these operators, and they derive sufficient conditions for a pooling operator to fully preserve the expressive power of the message-passing layers before it. This provides a universal and theoretically-grounded criterion for choosing among existing pooling operators or designing new ones. The exploration of unconventional pooling operators also adds to the originality of the work.

**Quality**: The quality of the paper is good in the rigorous theoretical analysis and comprehensive empirical evaluation. The authors align their empirical findings with their theoretical predictions well.

**Clarity**: The paper is organized and well-written. The theoretical analysis is clearly explained, and the presentation of the experimental results is easy to follow.

**Significance**: The findings in this paper contribute to a solid and necessary step in the understanding of pooling operators in GNNs. The theoretical conditions they propose for preserving the expressive power of message-passing layers and the new dataset could guide the design of new pooling operators.

**Weaknesses:**

- Since the proposed EXPWL1 dataset serves as an important contribution to the paper, the authors could provide more information about the dataset, i.e. at least provide certain example graph pairs.

- The experiments are kind of insufficient. First, the results in Table 1 are conducted under limited conditions with a pool ratio = 0.1 (excluding the pool ratio = 0.5 for Graclus and ECPool due to issues in implementation). It would be more comprehensive to see and discuss how the pooling operators compare to each other under different pool ratios. Second, the results in Table 3 in supplementary material do not contain the ones for Rand-dense, Cmp-Graclus, and Rand-sparse pooling operators. The comparison to these operators is necessary since the authors want to disprove the argument in [1] that such operators are comparable to the regular ones. Currently, the comparison is only done on a synthetic dataset.


[1] D. Mesquita, A. Souza, and S. Kaski. Rethinking pooling in graph neural networks. Advances in Neural Information Processing Systems, 33:2220–2231, 2020.

**Questions:**

- I'm wondering why only choosing a pooling ratio of 0.1 in the experiment. Would using larger pooling ratios affect the alignment between theoretical analysis and empirical findings? If using larger pooling ratios (e.g. 0.5), it is possible to compare the pooling operators more fairly in Table 1. Besides, a larger pooling ratio that doesn't reduce the graph size so aggressively can be possibly applied on the EXP dataset as an extension.

- For the claim in section 3.3: "Notably, this might not affect certain classification tasks, e.g., when the goal is to detect small structures that are already captured by the MP layers before pooling.", are there deeper investigations to support this when analyzing the empirical results?

**Limitations:**

The authors have not explicitly discussed the potential negative societal impacts of their work, which is not directly applicable.

---

> ### Author Rebuttal · Authors · 2023-08-04
>
> Thanks for your valuable feedback. We believe to have addressed all the reviewer's concerns and requests, as discussed in the following.
>
> > Provide examples of EXPWL1.
>
> We inserted in the supplementary material figures showing pairs of WL-1 distinguishable graphs from the EXPWL1 dataset. A few examples are in the PDF uploaded as part of the rebuttal.
>
> > Compare the operators using different pool ratios.
>
> We followed the suggestion of the reviewer and performed additional experiments using pooling ratios of 0.05 and 0.2 (see Table 2 in the PDF attached to the rebuttal).
> For higher pooling ratios the results do not change significantly for the expressive pooling methods, while we notice a drastic improvement in the performance of the non-expressive ones. These findings are aligned with our theoretical results since the sensitivity to different pooling ratios highlights the practical difference between expressive and non-expressive operators.
>
> Pheraps more importantly, after optimizing the code for memory usage, we managed to achieve a pooling ratio of approximately 0.2 with ECPool and 0.1 with every other method, making the comparison direct and fair.
>
> > Larger pooling ratios (e.g., 0.5).
>
> A ratio of 0.5 is not suitable for EXPWL1. As we discuss in the second remark of the experiments (section 4.3), 2 GIN layers are powerful enough to embed into **a single node** enough information to achieve a 92.1 accuracy on EXPWL1. Clearly, with a pooling ratio of 0.5 every method can embed the information in half of the nodes, achieving almost perfect accuracy no matter how such half is chosen. For this reason, a high ratio of 0.5 is not appropriate for comparing the pooling methods on EXPWL1.
>
> > Results for Rand-dense, Cmp-Graclus, and Rand-sparse in Tab. 3.
>
> We added the requested comparisons. See Table 5 in the PDF attached.
>
> > Consider the EXP dataset as an extension.
>
> The EXP dataset contains graphs that are not WL-1 distinguishable and is specifically designed for GNN architectures more powerful than the WL-1 test. Those go beyond the theoretical analysis done in this work and they are not designed to work with the hierarchical pooling layers analyzed.
>
> > Support the claim in section 3.3.
>
> To support our claim we added a reference to PGExplainer[1], which shows tasks where is sufficient to identify a small motif in the graph to correctly determine its class.
> On these tasks, a sparse method such as Topk can perform well by keeping only the part of the graph containing the motif of interest.
>
> [1] Luo, Dongsheng, et al. "Parameterized explainer for graph neural network." Advances in neural information processing systems 33 (2020): 19620-19631.

---

> > ### Comment · Reviewer_DheR · 2023-08-21
> >
> > I thank the authors for providing feedback and more results to address my concerns and requests. I have read other reviews and discussions. With all the changes to be implemented, I increase my score to 6. It would be also great if the authors can address the newly added results for Rand-Sparse, Rand-Dense, and Cmp-Graclus to support the claim in section 3.3.

---

### Official Review · Reviewer_utJZ · 2023-07-31

**Soundness:** 2 fair
**Presentation:** 2 fair
**Contribution:** 2 fair
**Rating:** 4
**Confidence:** 4

**Summary:**

Studies on the expressive power of Graph Neural Networks (GNNs) have garnered extensive attention. However, these studies have been limited to flat GNNs. Some hierarchical pooling methods, such as diff-pool, have been proposed. Evaluating the ability of pooling operations directly is challenging, so the performance of downstream tasks is often used as a measure. However, this approach is overly empirical and susceptible to other factors.

In this paper, the authors propose to use the ability to retain graph information to measure the power of pooling operations. Specifically, this paper first provides sufﬁcient conditions for a pooling operator to fully preserve the expressive power of the MP layers before it. Then, it reviews, classifies existing pooling operators.


**Strengths:**

1. This paper first studied on the expressive power of graph pooling in graph neural networks.

2. This paper categorizes the existing pooling methods based on whether they preserve the expressive ability of the Message Passing (MP) layer.


**Weaknesses:**

1. Given that simple global sum pooling can attain the same expressive capability as 1-WL (e.g., GIN), the rationale behind employing a complex pooling method that achieves no gain in expressive ability might be questioned.


2. The research lacks adequate experimentation. Ultimately, the investigation into enhancing or maintaining the expressive ability of Graph Neural Networks (GNNs) is driven by their impact on downstream tasks. However, the paper's classification performance is only evaluated on an extremely small dataset, which fails to provide sufficient convincing evidence.

3. The presentation requires improvement. For instance, reporting averages on different types of datasets in graph classification experiments, as shown in Figure 3, seems unusual and could be better structured.

4. There exists a disparity between Condition 1 of Theorem 1 and the corresponding explanation in line 137, i.e., $\mathcal{X}_1^L \neq \mathcal{X}_2^L $. The summation of the sets is not equal, is not equivalent to the formula in line 137.


**Questions:**

Could you please clarify the relationship between preserving all information after pooling, as defined in this paper, and the ability to distinguish graphs, which is the common definition of expressiveness? In this paper, the diff-pool aims to retain all information. However, from my perspective, the clustering process might lead to a loss of the graph's inherent structural information, which aligns with the description provided in line 244  (arguments against the evaluation approach in Paper [27]).

**Limitations:**

yes

---

> ### Author Rebuttal · Authors · 2023-08-04
>
> We thank the reviewer for the feedback. There are important misunderstandings that, hopefully, are clarified in the following.
>
> > 1. The rationale of hierarchical pooling vs. simpler global pooling.
>
> The rationale for using hierarchical pooling rather than global pooling is not to improve the expressive power, but to generate local summaries of the nodes, to gradually distill global information, and, above all, to enable further MP operations. Applying MP on a coarsened graph has a series of benefits, such as facilitating the exchange of information between distant parts of the graph.
>
> Arguing for the effectiveness of hierarchical pooling is not the focus of our work since it is extensively discussed and demonstrated in the original papers proposing the different pooling methods that we analyze.
>
> > 2. The classification performance is only evaluated on an extremely small dataset, which fails to provide sufficient convincing evidence.
>
> With due respect, we find this critique non-factual and incorrect:
>
> - The difference in the results obtained on EXPWL1 is statistically significant, so **there is** sufficient and convincing evidence to reject the null hypothesis that all pooling methods perform equally (the $p$-value is less than 0.001 for several differences of population means).
> - Is not true that we evaluate the performance **only on one dataset**. In fact, we consider 7 other datasets (plus 4 added in this rebuttal) that contain up to 10,000 graphs.
> - Since the size of EXPWL1 is aligned with common benchmark datasets for graph classifications, calling it **extremely small** seems inappropriate. Most importantly, being a synthetic dataset the size of EXPWL1 can be easily increased but it seemed unnecessary as, by doing that, we did not observe significant changes in the results.
>
> > 3. The presentation requires improvement. For instance, reporting averages on different types of datasets in graph classification experiments, as shown in Figure 3, seems unusual and could be better structured.
>
> Please, note that the details of each experiment are in the supplementary material.
>
> Since we performed **a lot** of experiments, we needed a way to summarize the results in a synthetic yet meaningful way. The purpose of Fig. 3 is to give an overview of the overall performances. To improve the presentation, we added the error bars in Fig. 3 (see attached PDF). We are happy to hear suggestions to further improve the presentation of the results.
>
> > 4. There exists a disparity between Condition 1 of Theorem 1 and the corresponding explanation in line 137, i.e., $X_1^L \neq X_2^L$. The summation of the sets is not equivalent to the formula in line 137.
>
> When using GIN layers, both Condition 1 of Theorem 1 and $X_1^L \neq X_2^L$ are satisfied.  We have made this clarification in the paper.
>
> > Could you please clarify the relationship between preserving all information after pooling, as defined in this paper, and the ability to distinguish graphs, which is the common definition of expressiveness?
>
> An expressive MP layer distinguishes two graphs by generating node embeddings that are distinct. If a pooling layer preserves all information the embeddings remain distinct and, consequently, the graphs remain distinguishable.
>
> > DiffPool keeps all information but the clustering process might lead to a loss of the graph's inherent structural information, which aligns with the description provided in line 244 (arguments against the evaluation approach in Paper [27]).
>
> One thing is to preserve the information in the node embeddings $\mathcal{X}^L$, another is to generate a coarsened graph with a structure compatible with the inductive bias of the next MP layer (e.g., nodes with similar features are connected). The latter task is deferred to the CONNECT function that, however, does not contribute to the definition of expressiveness and, thus, is not part of our theorem.
>
> An approach to estimate the capability of a pooling operator to preserve the structure of the original graph in the coarsened graph has been proposed in the SRC paper [1]. However, as we discuss in the introduction, this performance measure is often not aligned with the actual classification accuracy.
>
> [1] Grattarola, Daniele, et al. "Understanding pooling in graph neural networks." IEEE Transactions on Neural Networks and Learning Systems (2022).

---

> > ### Comment · Reviewer_utJZ · 2023-08-18
> > **Response to Authors**
> >
> > 1. I have reread the original paper and reviewed the author's rebuttal, and I will maintain my score.
> >
> > 2. My apologies for the typo in the second line of the initial review (a dataset). What I intended to convey is that the datasets used in this paper appear to be relatively small in scale. It seems that authors overlook prominent datasets such as OGB, zinc, and others that are currently of great interest in the field of GNN expressiveness.
> >
> > 3. In my opinion, it would be more intuitive and effective to include a column of average values in Table 3 of Appendix B3 as the result, rather than the results presented in Figure 3.
> >
> > 4. The formula in line 128 lacks clarity. The use of the same subscript for both $x$ and the summation symbol is unconventional and may cause confusion.

---

> > > ### Author Response · Authors · 2023-08-19
> > >
> > > Thanks for your answer.
> > >
> > > 2. The number of graphs in the dataset is not related in any way to the expressive power of a pooling operator. Therefore, regardless of the results obtained on OGB and zinc, the conclusion of our work would not change.
> > > The key element in our experimental evaluation is the result on EXPWL1, which empirically proves our theoretical conjecture. The purpose of testing real-world datasets is to show the utility of our theory in practice. We currently use **13** datasets to prove this second point. Unless there are datasets of graphs with special properties that could disprove our conclusion we see no reason to add yet another experiment.
> > >
> > > 3. Thanks for your opinion. It seems a matter of different tastes and it might be difficult to find an agreement. Our intention was to separate the experiment on EXPWL1 from the others, due to its importance and difference in nature. In our opinion, the figure contrasts well the accuracies vs. times and reflects better the qualitative nature of the second experiment on benchmark datasets.
> > >
> > > 4. We don't understand the issue: $\sum_i x_i$ is a standard notation for summation.

---

### Author Rebuttal · Authors · 2023-08-08

We thank the reviewers for recognizing the value of our work and for the useful feedback, which allowed to improve our paper.

Summary of changes:

- We addressed all the reviewers' requests by introducing explanations and several modifications in the text and the figures.
- We added an experiment with different pooling ratios.
- After a code optimization, we managed to achieve on EXPWL1 a pooling ratio of 0.2 with ECPool and 0.1 with every other method.
- We performed experiments on 6 new datasets: MUTAG, PTC-MR, IMDB-BINARY, IMDB-MULTI, ENZYMES, and REDDIT-MULTI-5K.
- We added results for Rand-Sparse, Rand-Dense, and Cmp-Graclus on every benchmark dataset.
- We added results on EXPWL1 with a GCN baseline.

---

### Decision · Program_Chairs · 2023-09-21

**Decision:**

Accept (poster)

**Comment:**

This paper delves into the expressive capabilities of pooling operations in Graph Neural Networks (GNNs). The authors propose specific conditions for a pooling operator to fully retain the expressive capability of preceding MP layers and classify existing operators based on this criteria. The paper integrates both theoretical and empirical analyses and introduces a new dataset, EXPWL1, to support their claims. While the study holds promise, there are concerns regarding the comprehensiveness of experiments, presentation clarity, and the theoretical context in which the results are framed. The major concern lies on the limited experimentation. The datasets used for evaluating classification performance are reportedly small, raising questions about the robustness of the results. Thus, the authors are strongly encouraged to compare different pooling operators on larger datasets such as OGB and zinc, and also with vanilla sum/mean pooling, to broaden the impact of the work.